# Factoring an integer with three oscillators and a qubit

Lukas Brenner [1,2,6] ✉, Libor Caha [1,2,6] ✉, Xavier Coiteux-Roy [1,2,3,4,5,6] ✉ & Robert Koenig [1,2,6] ✉

A common starting point of traditional quantum algorithm design is the notion of a universal quantum computer with a scalable number of qubits. This convenient abstraction mirrors classical computations manipulating bits. It allows for a device-independent development of algorithmic primitives. Here we argue that an alternative approach centered on the physical setup can yield great benefits. As an example, we consider hybrid qubit-oscillator systems with linear optics operations augmented by certain qubit-controlled Gaussian unitaries. The continuous variable Fourier transform and certain arithmetic operations have native realizations in such systems. We put this to algorithmic use and give a polynomial-time quantum factoring algorithm which uses only one qubit and three oscillators, independent of the number being factored.

The notion of a universal, scalable quantum computer as succinctly formulated by DiVincenzo's criteria[1] stipulates that a computationally useful quantum device needs to provide a number of (logical) qubits scaling extensively with the problem size. Following this idea, the most established current approach to quantum computing with continuous-variable (CV) systems (that have a long history, see e.g., refs. 2–9) is to encode a single logical qubit in a suitable two-dimensional subspace of a bosonic mode. By associating a single physical information carrier (an oscillator) to each individual qubit, this philosophy emphasizes modularity, breaking the engineering challenge into more manageable pieces, and allowing for device-independent approaches in algorithms design. It amounts to a quite literal interpretation of what it means to scale a quantum computation.

Here, we argue that in light of the possibilities offered by CV quantum systems, this simple and seemingly inevitable idea of scalability may be too restrictive. Specifically, consider Shor's integer factoring algorithm[10]: Following the standard paradigm, it requires a number of qubits that is proportional to the number of bits specifying the integer to be factored. With our alternative approach, we can show that, instead, only three oscillators and a qubit suffice to efficiently factor any arbitrarily large integer $N$: We give a quantum algorithm which factors an $n$-bit integer (for any

integer $n$) using a polynomial number of elementary operations which are readily available in present-day experimental setups. In other words, our algorithm trades problem size (length $n$ of the binary representation of the integer to be factored) against circuit size (i.e., number of gates) while keeping the underlying physical system (three oscillators and one qubit) fixed. Since any finite-dimensional space can trivially be embedded into a single harmonic oscillator−the question of how to realize a computation with a small number of CV information carriers is only meaningful when restricting to basic, physically realizable operations, and with an estimate on the number of operations used.

## Results

### The qubit-oscillator model of computation

The set of elementary operations we consider is a subset of the toolbox available in hybrid qubit-oscillator systems, see ref. 11 for an up-to-date review including a discussion of physical realizations. Concretely, we count as one elementary operation each of the following:

(i) Preparation of the computational basis state $|0\rangle$ of the qubit, and of the single-mode vacuum state $|\mathrm{vac}\rangle$ on any of the three modes.

[1]School of Computation, Information and Technology, Technical University of Munich, Munich, Germany. [2]Munich Center for Quantum Science and Technology, Munich, Germany. [3]School of Natural Sciences, Technical University of Munich, Munich, Germany. [4]Department of Computer Science, University of Calgary, Calgary, AB, Canada. [5]Department of Physics and Astronomy, University of Calgary, Calgary, AB, Canada. [6]These authors contributed equally: Lukas Brenner, Libor Caha, Xavier Coiteux-Roy, Robert Koenig. ✉e-mail: lukas.brenner@tum.de; cahalibor@me.com; xavier.coiteuxroy@ucalgary.ca; robert.koenig@tum.de

(ii)  Clifford gates on the qubit, and single-mode phase space rotations, translations, squeezing, as well as two-mode beam splitters (on any pair of modes), with parameters (such as angles) bounded by a constant.

(iii) Qubit-controlled phase space displacements, and qubit-controlled phase space rotations, with parameters bounded by a constant.

(iv) Computational basis measurement on the qubit and homodyne quadrature measurements on the bosonic modes.

The unitaries (iii) are generated by Jaynes-Cummings-type Hamiltonians (see e.g., refs. [12–14]) and enable the realization of non-Gaussian operations on the oscillators. An example is the preparation of approximate, i.e., finitely squeezed, Gottesman-Kitaev-Preskill (GKP) states[15], defined as follows (for convenience, our convention differs slightly from the error-correction literature, where peaks are typically centered on integer multiples of $\sqrt{2\pi}$ instead of integers):

$$\mathrm{GKP}_{\kappa,\Delta}(x) \propto \sum_{z\in\mathbb{Z}} e^{-\kappa^2 z^2/2} e^{-(x-z)^2/(2\Delta^2)} \qquad \text{for} \qquad x\in\mathbb{R},$$

where $\kappa, \Delta > 0$. In ref. [16], we give a protocol $\mathcal{P}_{\kappa,\Delta}^{\mathrm{GKP}}$ achieving this with a constant success probability (which can be amplified by repetition to

an arbitrary constant without change in complexity), polynomial error in trace distance in $(\kappa, \Delta)$, and a number of elementary operations which is linear in $(\log 1/\kappa, \log 1/\Delta)$, see Theorem 2.2 in the Supplementary Note for details.

### Description of the factoring algorithm

The elementary operations (ii), (iii) can be used to realize certain (real) arithmetic operations when the action on bosonic position-eigenstates $|x\rangle$, $x\in\mathbb{R}$ and qubit computational basis states $|b\rangle$, $b \in \{0, 1\}$ is considered, see Fig. 1. Our algorithm relies on an extended arithmetic toolbox associated with certain composite unitaries. These realize specific arithmetic functionalities, see Fig. 2. One of these unitaries coherently performs a form of modular exponentiation: It computes $x \mapsto f_{a,N,m}(x)$ for a function $f_{a,N,m}$ such that

$$f_{a,N,m}(x) \equiv a^x \,(\mathrm{mod}\ N) \qquad \text{for all} \qquad x\in\{0,\dots,2^m-1\},$$

such a function can be understood as a proxy for the modular exponentiation map $x \mapsto a^x \bmod N$ (with $a, x \in \mathbb{N}$). Correspondingly, we refer to $f_{a,N,m}$ as a pseudomodular power. (In our algorithm, we set $m$ to be proportional to $n$, with a constant to be fixed later, see the Supplementary Table 1).

| One-/two-mode Gaussian operation | Circuit representation | Definition |
|---|---|---|
| $(Q$-)displacement by $\pm 1$ | $\lvert x\rangle \ \boxed{e^{\mp iP}}\ \lvert x \mp 1\rangle$ | $e^{\mp iP}$ |
| Bosonic sum | $\lvert x\rangle \rightarrow \lvert x\rangle$ ; $\lvert y\rangle \oplus \lvert x+y\rangle$ | $e^{-iQ_1 P_2}$ |
| Single-mode squeezing | $\lvert x\rangle \ \boxed{S(z)}\ e^{z/2}\lvert e^{-z}\cdot x\rangle$ | $e^{iz(QP+PQ)/2}$ |
| $(P$-)quadrature measurement | $\lvert\psi\rangle \ \measuredangle_P\ p$ | $p \sim \lvert\widehat{\psi}(p)\rvert^2 dp$ |

| Qubit-controlled Gaussian unitary | Circuit representation | Definition |
|---|---|---|
| $(Q$-)displacement by $1$ | $\lvert b\rangle \rightarrow \lvert b\rangle$ ; $\lvert x\rangle \oplus \lvert x+b\rangle$ | $\lvert 0\rangle\langle 0\rvert \otimes I + \lvert 1\rangle\langle 1\rvert \otimes e^{-iP}$ |
| $(Q$-)displacement by $-1$ | $\lvert b\rangle \rightarrow \lvert b\rangle$ ; $\lvert x\rangle \ominus \lvert x-b\rangle$ | $\lvert 0\rangle\langle 0\rvert \otimes I + \lvert 1\rangle\langle 1\rvert \otimes e^{iP}$ |
| $(P$-)displacement by $\pm\pi$ | $\lvert b\rangle \rightarrow \lvert b\rangle$ ; $\lvert x\rangle \ \boxed{e^{\pm i\pi Q}}\ e^{\pm i\pi xb}\lvert x\rangle$ | $\lvert 0\rangle\langle 0\rvert \otimes I + \lvert 1\rangle\langle 1\rvert \otimes e^{\pm i\pi Q}$ |
| Phase-space rotation by $\pm\pi/2$ | $\lvert b\rangle \rightarrow \lvert b\rangle$ ; $\lvert\psi\rangle \ \boxed{e^{\pm i\pi\widehat{N}/2}}\ e^{\pm i\pi b\widehat{N}/2}\lvert\psi\rangle$ | $\lvert 0\rangle\langle 0\rvert \otimes I + \lvert 1\rangle\langle 1\rvert \otimes e^{\pm i\pi\widehat{N}/2}$ |

**Fig. 1 | Elementary operations.** Circuit representations illustrating the action on position-eigenstates $\{\lvert x\rangle\}_{x\in\mathbb{R}}$ of bosonic modes (thick wires) and computational basis states $\{\lvert b\rangle\}_{b\in\{0,1\}}$ of a qubit (thin gray wires). The two-mode bosonic addition gate $e^{-iQ_1 P_2}$ can be decomposed into constantly many beam-splitters and single mode squeezing unitaries, see refs. [17,18]. A homodyne $P$-quadrature measurement applied to a state $\lvert\Psi\rangle \in L^2(\mathbb{R})$ produces a sample $p \in \mathbb{R}$ from the distribution with density function $p \mapsto \lvert\widehat{\Psi}(p)\rvert^2$, where $\widehat{\Psi}(p) := \frac{1}{(2\pi)^{1/2}}\int \Psi(x)e^{ipx}dx$ denotes the Fourier transform of $\Psi$. Homodyne $Q$-quadrature measurement is defined similarly. Controlled-phase space rotations are defined in terms of the number operator $\widehat{N} = (Q^2 + P^2 - I)/2$.

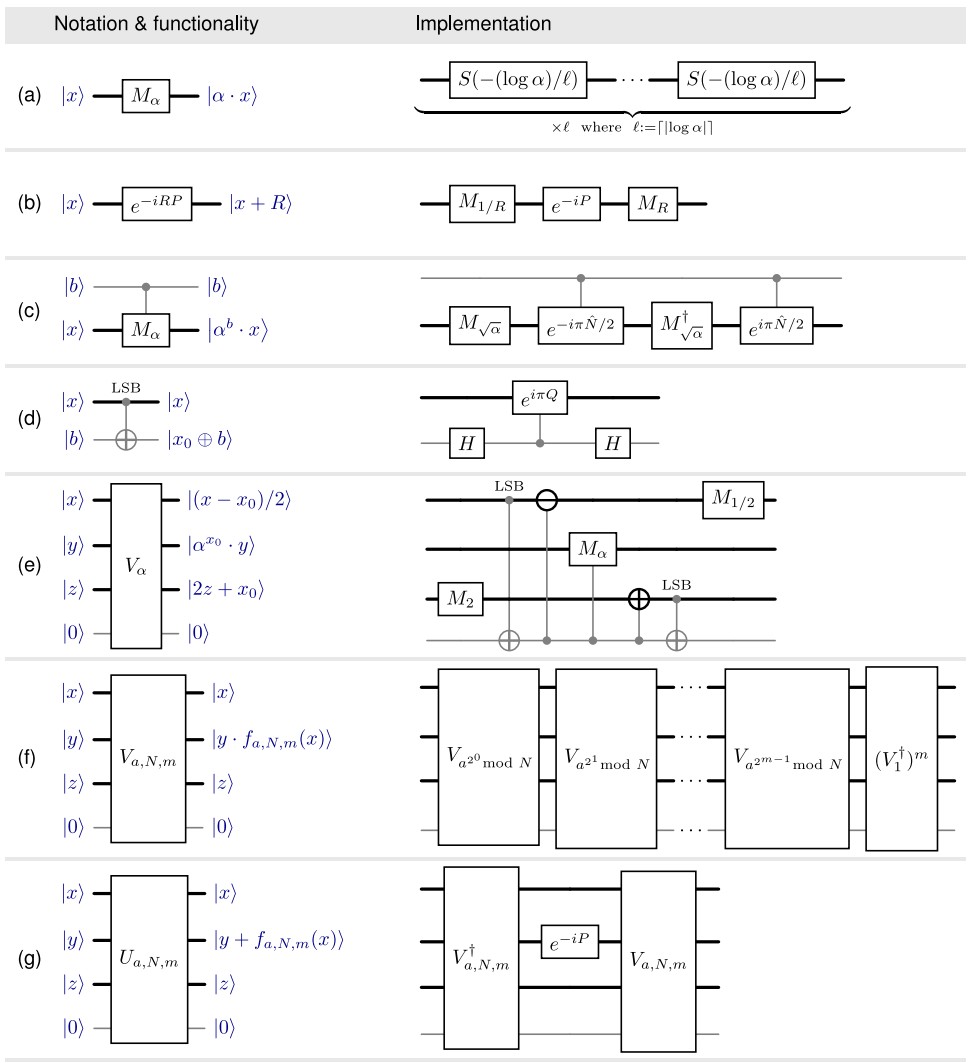

**Fig. 2 | Composite unitaries.** Actions are specified in blue (suppressing normalization factors) for $x = \sum_{i=0}^{m-1} 2^i x_i$ an $m$-bit integer, $b \in \{0, 1\}$, $z \in \mathbb{N}_0$, $y \in \mathbb{R}$ and $a, N, m \in \mathbb{N}$. They are: **a** scalar multiplication by a real number $\alpha > 0$ (realized by $\ell$ constant-strength squeezing operations), **b** translation by $R > 0$, **c** qubit-controlled scalar multiplication, **d** coherent extraction of the least significant bit (LSB) $x_0$ of $x$, and **e** an auxiliary unitary $V_\alpha$ realizing multiplication of the position $y$ of the second mode by $\alpha > 0$, controlled on the LSB $x_0$ of the position $x$ of the first mode. This unitary moves the bit $x_0$ to the third auxiliary mode and makes $x_1$ the new LSB of the position in the first mode. This is used to build (**f**) the unitary $V_{a,N,m}$ which multiplies the position of the second mode by the pseudomodular power $f_{a,N,m}(x) = \prod_{i=0}^{m-1} \left( a^{2^i} \bmod N \right)^{x_i}$ of the position $x$ of the first mode. Finally, **g** the unitary $U_{a,N,m}$ implementing a translation by the pseudomodular power $f_{a,N,m}(x)$ of the second bosonic mode controlled by the position $x$ of the first mode.

With these preparations, we can complete the description of our algorithm. Its quantum subroutine is given by the circuit $\mathcal{Q}_{a,N}$ depicted in Fig. 3, but with the approximate initial GKP states $|GKP_{\kappa_A, \Delta_A}\rangle$, $|GKP_{\kappa_B, \Delta_B}\rangle$ replaced by the output states of the preparation procedure $\mathcal{P}^{GKP}_{\kappa_A, \Delta_A}$ and $\mathcal{P}^{GKP}_{\kappa_B, \Delta_B}$ (for suitably chosen parameters $\kappa_A$, $\Delta_A$, $\kappa_B$, $\Delta_B$), respectively. We note that for our choice of parameters (see Supplementary Table 1), both the circuit $\mathcal{Q}_{a,N}$ and the preparation procedure $\mathcal{P}^{GKP}_{\kappa, \Delta}$, and hence also our quantum subroutine, use $O(n^2)$ elementary operations (i)–(iv).

Assume that we run the circuit $\mathcal{Q}_{a,N}$ with initial GKP states $|GKP_{\kappa_A, \Delta_A}\rangle$, $|GKP_{\kappa_B, \Delta_B}\rangle$. Key to our algorithm is the fact that a single sample from the output distribution of this circuit can be postprocessed by an efficient, i.e., polynomial-time classical algorithm, yielding a factor of $N$ with a substantial probability. We have the following:

**Lemma 1.** Suppose $N$ is an $n$-bit number. There is a polynomial-time classical algorithm which—given a uniformly chosen element $a \sim \mathbb{Z}_N^*$

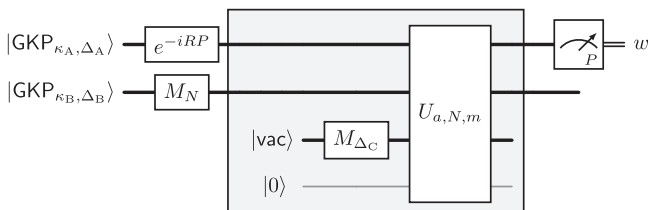

**Fig. 3 | The physically realizable quantum circuit $\mathcal{Q}_{a,N}$, where $a \in \mathbb{Z}_N^*$.** See Supplementary Table 1 for a suitable choice of parameters $R, m, \Delta_A, \kappa_A, \Delta_B, \kappa_B, \Delta_C$. It applies a sequence of elementary operations and derived unitaries (see Figs. 1, 2) to two approximate GKP states in the first and second mode, a vacuum state $|vac\rangle$ in the third mode and a qubit computational basis state $|0\rangle$ in the qubit system. The output is a sample $w \in \mathbb{R}$ obtained by performing a $P$-quadrature measurement on the first mode. The shaded subcircuit approximately computes a pseudomodular power. To provide intuition, we will first discuss the effect of the circuit when this subcircuit is replaced by an ideal unitary $U^{ideal}_{\mathbb{R}, a, N}$ computing the real power, see Fig. 4b.

(a) Shor's circuit $\mathcal{Q}_{a,N}^{\text{Shor}}$.

(b) An idealized hybrid circuit $\mathcal{Q}_{a,N}^{\text{ideal}}$.

**Fig. 4 | Contrasting Shor's circuit with an idealized hybrid circuit $\mathcal{Q}_{a,N}^{\text{ideal}}$.** Instead of modular exponentiation followed by Fourier transform and computational-basis measurement in Shor's circuit (**a**), our idealized hybrid circuit $\mathcal{Q}_{a,N}^{\text{ideal}}$ (**b**) employs standard (real) exponentiation and homodyne detection; modularity (i.e., realizing (mod $N$) computations) is recovered by initializing the second bosonic mode in a modified GKP state with spacing $N$.

and a single sample from the output distribution of the circuit $\mathcal{Q}_{a,N}$ (see Fig. 3)—produces a factor of $N$ with probability at least $\Omega(1/\log n)$.

Furthermore, replacing the initial approximate GKP states with the states prepared by the preparation protocol $\mathcal{P}_{\kappa,\Delta}^{\text{GKP}}$ in circuit $\mathcal{Q}_{a,N}$ leads to a statement comparable to Lemma 1. By repetition with a randomly chosen $a \in \mathbb{Z}_N^*$, we obtain the following by a suitable choice of parameters (see Supplementary Table 1):

**Theorem 1.** (Efficient quantum algorithm for factoring) There is a polynomial-time algorithm which, given an $n$-bit integer $N$,

(I) Repeatedly uses a quantum circuit on three oscillators and one qubit consisting of $O(n^2)$ elementary operations (i)–(iv), and

(II) produces a factor of $N$ with constant probability.

(The success probability can be amplified by repetition or other techniques as in ref. 10).

Our factoring algorithm is the result of translating Shor's algorithm[10] to CV systems, with specific modifications exploiting their potential. These modifications mean that our construction—although formally quite similar to Shor's algorithm—is not simply obtained by embedding a finite-dimensional quantum computation into an infinite-dimensional system. Instead, our approach relies on different algebraic structures: We use an approximate GKP state as a proxy for a uniform superposition over all integers instead of a uniform superposition over all $n$-bit integers as in Shor's algorithm. To our knowledge, such an algorithmic use of approximate GKP states is new. Instead of realizing modular arithmetic by gates acting on finite-dimensional systems, our gates natively perform real arithmetic. To realize modular arithmetic, we exploit the (approximate) stabilization property of (approximate) GKP states under suitable (discrete) displacements, a property that underlies their relevance for quantum error correction. Finally, our algorithm leverages the Fourier transform on $\mathbb{R}^n$ instead of the Fourier transform over a finite cyclic group. With these alternative choices, we can show that the physical operations (i)–(iv) (together with an efficient classical computation) are sufficient to address the factoring problem. The difference between our approach and Shor's algorithm is further expressed in the different complexities: the quantum subroutine of our algorithm has circuit size $O(n^2)$, whereas that of Shor's algorithm has size $O(n^2 \log n)$, when using the best currently known classical algorithm for multiplication with time complexity $O(n \log n)$ of ref. 19.

As discussed further in the "Discussion" section, our algorithm produces states whose energy scales extensively in $N$ (the number being factored). This is a natural consequence of an embedding of an exponentially large Hilbert space into a system of three oscillators and one qubit.

## Key ideas

To motivate our construction, recall the basic ingredients of Shor's algorithm[10], which relies on modular arithmetic and the Fourier transform over $\mathbb{Z}_q$. It builds on the (randomized) reduction[20] of the problem of finding a prime factor of an (odd) integer $N$ to the problem of finding the period $r$ of the function $f_{a,N}(x) = a^x \bmod N$, where $a \in \mathbb{Z}_N^*$. (The period $r$ of $f_{a,N}$ can be used to compute a factor of $N$ whenever $r$ is even and $x^{r/2} \neq -1 \bmod N$. For a uniformly random integer $a \in \mathbb{Z}_N^*$, the necessary conditions on the period $r$ of $f_{a,N}$ hold except with probability $2^{1-k}$, where $k$ is the number of distinct prime factors of $N$, a bound derived from the Chinese remainder theorem.) Given a pair $(a, N)$, Shor's algorithm applies an efficient classical post-processing algorithm to a sample $c$ produced by a (polynomial-size) quantum circuit $\mathcal{Q}_{a,N}^{\text{Shor}}$. This gives the period $r$ of the function $f_{a,N}$ with probability at least $\Omega(\frac{1}{\log \log r})$, resulting in a polynomial-time factorization algorithm by repeated application.

A high-level description of Shor's circuit $\mathcal{Q}_{a,N}^{\text{Shor}}$ is given in Fig. 4a. It uses a binary encoding of $(\log_2 q)$-bit integers into $\log_2 q$ qubits, where $q$ is the smallest power of 2 such that $N^2 < q$. The circuit starts with the first register in the uniform superposition $\frac{1}{q^{1/2}} \sum_{x=0}^{q-1} |x\rangle$ over all integers $x \in \{0, \ldots, q-1\} =: \mathbb{Z}_q$, coherently computes the function $f_{a,N}$ into a second register using a modular exponentiation unitary $U_{\mathbb{Z}_N, a}$ acting as

$$U_{\mathbb{Z}_N, a}(|x\rangle \otimes |y\rangle) = |x\rangle \otimes |y + (a^x \bmod N)\rangle,$$

applies the (discrete) Fourier transform $\text{FT}_{\mathbb{Z}_q}$ on $L^2(\mathbb{Z}_q)$ and finally measures the first register in the computational basis giving an outcome $c \in \mathbb{Z}_q$. Shor shows that the output distribution $p_{\mathbb{Z}_q}(c)$ is close to the uniform distribution on the set $\{\frac{q}{r} \cdot d \mid d \in \{0, \ldots, r-1\}\}$, i.e., the output $c$ is an integer multiple of $q/r$ with high probability. This property enables the extraction of the period $r$ (The classical post-processing algorithm proceeds as follows: With the continued fraction expansion of $c/q$, the number $c/q$ is rounded to the nearest fraction of the form $d'/r'$ with denominator $r'$ smaller than $N$. The value of $r = r'$ can then be extracted whenever $d'$ and $r'$ are coprime).

Here we argue that—in place of the circuit $\mathcal{Q}_{a,N}^{\text{Shor}}$—the idealized hybrid circuit $\mathcal{Q}_{a,N}^{\text{ideal}}$ given in Fig. 4b can be used, assuming that the classical post-processing procedure is appropriately modified. We note that the circuit $\mathcal{Q}_{a,N}^{\text{ideal}}$ involves non-normalizable GKP states, i.e., the formal uniform superposition

$$|\text{GKP}\rangle \propto \sum_{x \in \mathbb{Z}} |x\rangle. \tag{1}$$

The circuit $\mathcal{Q}_{a,N}^{\text{ideal}}$ is thus not physical, but it nevertheless illustrates the key ideas underlying our physical circuit $\mathcal{Q}_{a,N}$ in Fig. 3. Let us highlight how it differs from Shor's circuit $\mathcal{Q}_{a,N}^{\text{Shor}}$.

First, our algorithm relies on real instead of modular arithmetic. Clearly, the expression (1) is a natural analog of the uniform superposition of basis states used in Shor's circuit $\mathcal{Q}_{a,N}^{\text{Shor}}$. The hybrid circuit $\mathcal{Q}_{a,N}^{\text{ideal}}$ also uses a CV-analog of the modular exponentiation unitary $U_{\mathbb{Z}_N, a}$. Concretely, consider the unitary $U_{\mathbb{R}, a, N}^{\text{ideal}}$ which acts on pairs of

**Fig. 5 | Modular position measurement using an auxiliary GKP state.** The modular value $x \bmod N$ of a position-eigenstate $|x\rangle$ is obtained by adding $x$ to an auxiliary system in the state $M_N|\mathrm{GKP}\rangle$, and then measuring the $Q$-quadrature of that system. Because of the fact that $M_N|\mathrm{GKP}\rangle \propto \sum_{y\in\mathbb{Z}}|y\cdot N\rangle$ is the uniform superposition of position-eigenstates with spacing $N$, this circuit provides $w \equiv x \pmod N$ (and no other information on $x$). This reasoning has been used to construct syndrome extraction circuits for GKP-codes[15].

position-eigenstates $|x\rangle$, $|y\rangle$ as

$$U^{\mathrm{ideal}}_{\mathbb{R},a,N}(|x\rangle\otimes|y\rangle)=|x\rangle\otimes|y+a^x\rangle \quad\text{where}\quad a^x := \begin{cases} a^x & \text{if } x\geq 0 \\ (a^{-1}\bmod N)^{|x|} & \text{if } x<0 \end{cases}. \tag{2}$$

Importantly, the exponentiation in the definition of $a^x$ is not taken modulo $N$, but is to be understood over the non-negative reals. Integer values belonging to $\mathbb{Z}_N$ are only obtained subsequently by means of a modular measurement.

As already mentioned, our hybrid circuit $\mathcal{Q}^{\mathrm{ideal}}_{a,N}$ exploits a peculiarity of CV-systems that allows to circumvent the need for constructing an implementation of the unitary Fourier transform. Indeed, homodyne $P$-quadrature measurements (natively available in typical quantum-optical systems) are equivalent to a Fourier transform $\mathrm{FT}_{\mathbb{R}}$ on $L^2(\mathbb{R})$ followed by a homodyne $Q$-quadrature measurement. Motivated by this, the circuit $\mathcal{Q}^{\mathrm{ideal}}_{a,N}$ simply applies a homodyne $P$-quadrature measurement on the first mode, giving a sample from the distribution

$$p'_{\mathbb{R}}(w) \propto \left\| (\langle\hat{w}|\otimes I)\, U^{\mathrm{ideal}}_{\mathbb{R},a,N}(|\mathrm{GKP}\rangle\otimes M_N|\mathrm{GKP}\rangle) \right\|^2, \tag{3}$$

for $w\in\mathbb{R}$. Here $|\hat{w}\rangle := \mathrm{FT}_{\mathbb{R}}|w\rangle$ is the momentum-eigenstate to value $w$, i.e., $P|\hat{w}\rangle = w|\hat{w}\rangle$.

A further key difference between Shor's circuit $\mathcal{Q}^{\mathrm{Shor}}_{a,N}$ and our hybrid circuit $\mathcal{Q}^{\mathrm{ideal}}_{a,N}$ is the choice of initial state in the second register (see Fig. 4): Whereas in Shor's circuit, the auxiliary register is initialized in the state $|0\rangle$ (corresponding to $0\in\mathbb{Z}_q$), the idealized circuit $\mathcal{Q}^{\mathrm{ideal}}_{a,N}$ uses $M_N|\mathrm{GKP}\rangle$ in its place. This is a uniform superposition $M_N|\mathrm{GKP}\rangle \propto \sum_{y\in\mathbb{Z}}|y\cdot N\rangle$ of integer multiples of $N$. This choice is dictated by the need to realize a computation modulo $N$ (rather than only real arithmetic): Together with the subsequent steps in the algorithm, it essentially realizes a modular $P$-quadrature measurement.

We note that modular measurements have traditionally been used in syndrome extraction circuits for GKP-codes[15] see Fig. 5. Our circuit $\mathcal{Q}^{\mathrm{ideal}}_{a,N}$ is partly motivated by such modular measurement circuits. Indeed, by combining real arithmetic with this approach, they effectively realize an analog of modular exponentiation followed by measurement.

A brief computation shows that the output distribution $p'_{\mathbb{R}}$ of the circuit $\mathcal{Q}^{\mathrm{ideal}}_{a,N}$ (see Eq. (3)) is (formally) the uniform distribution on the set $\{j/r \mid j\in\mathbb{Z}\}$. The period $r$ can be recovered immediately from $j/r$ whenever $j$ and $r$ are coprime. Following Shor's analysis, such a favorable outcome is obtained with probability at least $\Omega(1/\log\log r)$, leading to a polynomial runtime when the circuit is used repeatedly. (The integers $j$ and $r$ are coprime with probability at least $\phi(r)/r$, where $\phi(\cdot)$ is Euler's totient function. Using the bound given in ref. 21, Thm.328], one can conclude that we recover the period $r$ of $a$ modulo $N$ with probability at least $\delta/\log\log r$ for some $\delta = \Theta(1)$. Therefore, it is enough to repeat the procedure $\log\log N$ times, i.e., logarithmically

many times in the number of bits of $N$ to succeed with probability at least $1-e^{-1}$).

To show the claim that the output distribution $p'_{\mathbb{R}}$ is (formally) uniform on the set $\{j/r \mid j\in\mathbb{Z}\}$, first observe that the state before the measurement can be written as

$$U^{\mathrm{ideal}}_{\mathbb{R},a,N}(|\mathrm{GKP}\rangle\otimes M_N|\mathrm{GKP}\rangle) \propto \sum_{x,y\in\mathbb{Z}}|x\rangle\otimes|y\cdot N+a^x\rangle$$
$$\propto \sum_{x,y\in\mathbb{Z}}|x\rangle\otimes|y\cdot N+(a^x\bmod N)\rangle. \tag{4}$$

Modular arithmetic arises here because of the invariance of the state $M_N|\mathrm{GKP}\rangle$ under translations by integer multiples of $N$. The reduced density operator on the first mode of the state in Eq. (4) is a mixture of pure states, each of which has a period $r$ in position space. As a consequence, applying a homodyne momentum measurement to the first mode gives a uniformly chosen integer multiple of $1/r$ (see the Supplementary Note).

## A circuit composed of physical operations

This formal discussion of the idealized circuit $\mathcal{Q}^{\mathrm{ideal}}_{a,N}$ merely illustrates the basic ideas. The circuit $\mathcal{Q}^{\mathrm{ideal}}_{a,N}$ falls short of being a physically amenable in two important ways. First, it relies on idealized (infinitely squeezed) GKP states which are unnormalizable and hence unphysical. Second, we have not provided a circuit decomposition of the unitary $U^{\mathrm{ideal}}_{\mathbb{R},a,N}$ (defined in Eq. (2)) into elementary operations from the list (i)–(iv). These issues are addressed by the circuit $\mathcal{Q}_{a,N}$ (see Fig. 3), which uses as input two approximate GKP states $|\mathrm{GKP}_{\kappa_A,\Delta_A}\rangle$ and $|\mathrm{GKP}_{\kappa_B,\Delta_B}\rangle$ with parameters $\kappa_A = \Delta_A = 2^{-\Theta(n)}$ and $\kappa_B = \Delta_B = 2^{-\Theta(n^2)}$, respectively. The latter can be produced (approximately) by the protocol $\mathcal{P}^{\mathrm{GKP}}_{\kappa,\Delta}$. The unitaries $M_N$, $M_{\Delta_C}$ and $e^{-iRP}$ (where $\Delta_C = 2^{-\Theta(n)}$ and $R = 2^{\Theta(n)}$) can be realized by $O(n)$ elementary operations (see Fig. 2). The unitary $U_{a,N,m}$ on $L^2(\mathbb{R})^{\otimes 3}\otimes\mathbb{C}^2$, where $m = \Theta(n)$ (see Fig. 2) can be realized with $O(n^2)$ elementary operations.

The unitary $U_{a,N,m}$ computes the pseudo-modular power function $f_{a,N,m}$ when applied to a state where the position of the first mode is $x\in\{0,\ldots,2^m-1\}$, the position $y\in\mathbb{R}$ of the second mode is arbitrary, and the position of the third mode is $z=0$. That is, we have

$$U_{a,N,m}(|x\rangle\otimes|y\rangle\otimes|0\rangle\otimes|0\rangle)=|x\rangle\otimes|y+f_{a,N,m}(x)\rangle\otimes|0\rangle\otimes|0\rangle,$$

(Here the qubit is in the state $|0\rangle$.) The unitary $U_{a,N,m}$ takes the role of $U^{\mathrm{ideal}}_{\mathbb{R},a,N}(a)$ in the idealized circuit $\mathcal{Q}^{\mathrm{ideal}}_{a,N}$ (see Fig. 4b). To ensure that the position of the first mode has most of its support on the set $\{0,\ldots,2^m-1\}$, the circuit $\mathcal{Q}_{a,N}$ involves the unitary $e^{-iRP}$ which shifts the center of the initial state $|\mathrm{GKP}_{\kappa_A,\Delta_A}\rangle$ to the right. We give a decomposition of the unitary $U_{a,N,m}$ into elementary operations in the Supplementary Note. We also show this decomposition has the claimed complexity.

The use of physically realizable (approximate) GKP states $|\mathrm{GKP}_{\kappa_A,\Delta_A}\rangle$ and $|\mathrm{GKP}_{\kappa_B,\Delta_B}\rangle$ with (finite) squeezing parameters $(\kappa_A, \Delta_A)$ and $(\kappa_B, \Delta_B)$ in the circuit $\mathcal{Q}_{a,N}$ necessitates a detailed analysis of approximate function evaluation by unitaries on $L^2(\mathbb{R})^{\otimes 3}\otimes\mathbb{C}^2$. This analysis, given in the Supplementary Note, provides detailed estimates on the correctness of the algorithm as a function of the squeezing parameters. Specifically, we have to consider non-integer inputs, i.e., when positions $x\in\mathbb{R}\backslash\mathbb{Z}$ are involved. By continuity arguments and using the definition of approximate GKP states, we argue that the considered unitaries on inputs supported sufficiently close to the set of integers approximately implement a rounded version of function

evaluation. Applied to the circuit $\mathcal{Q}_{a,N}$, this analysis shows that the associated output distribution on $\mathbb{R}$ still has the property that the samples can be post-processed to find the period $r$ with a significant probability.

## Discussion

Our polynomial-time factoring algorithm exemplifies the benefits of hardware-aware quantum algorithms design: We show how to algorithmically leverage a set of natively available operations in hybrid qubit-oscillator-systems. This draws attention to a physically motivated computational model which deserves further study from a complexity-theoretic perspective.

Despite the use of physically realistic elementary operations and the efficiency guarantee we establish, our proposal is a proof-of-principle of theoretical nature only. While our algorithm identifies a new connection between CV quantum error-correcting codes (in the form of GKP states) and quantum algorithms, it does not incorporate fault-tolerance considerations. More significantly, even though our algorithm only uses a polynomial number of active single-mode (squeezing) operations, the states produced in the course of our algorithm have an energy growing extensively in the number to be factored. This means that factoring even a modestly-sized integer (such as $N = 21$, the largest number previously experimentally factored[22] with Shor's algorithm) using our approach will be challenging.

For these reasons, our algorithm should primarily be seen as an exploration of an abstract model of computation rather than an experimentally viable route towards factoring numbers of practically relevant size (such as those used in encryption schemes). In a similar vein, Shamir has proposed a (classical) algorithm[23] which finds a factor of $N$ with only $O(\log N)$ integer arithmetic operations. But unlike Shamir's algorithm—which presupposes a computational model allowing integer arithmetic operations at unit cost—our work centers around a concrete physical model describing readily available experimental setups[11]. In more recent work, Chabaud et al.[24] analyze the computational power of CV systems with superquadratic Hamiltonians. We emphasize that their model significantly differs from ours as individual gates can increase the energy by a superlinear amount. Such gates appear challenging to realize experimentally at present.

Similar to the way Shor's algorithm spurred interest in experimental platforms, quantum error-correction, quantum complexity theory and quantum algorithms, our work may motivate further research strengthening connections between theoretical computer science and quantum physics.

## Data availability

No datasets were generated or analyzed during the current study.

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

## Acknowledgements

L.B., L.C. and R.K. acknowledge support by the European Research Council under grant agreement no. 101001976 (project EQUIPTNT), as well as the Munich Quantum Valley, which is supported by the Bavarian state government through the Hightech Agenda Bayern Plus. X.C.R. thanks the Swiss National Science Foundation (SNSF) for its financial support and acknowledges funding from the BMW endowment fund.

## Author contributions

L.B., L.C., X.C.R. and R.K. contributed equally to this work.

## Funding

## Competing interests

The authors declare no competing interests.
