## [Transparent Peer Review file · Nature Communications]

Factoring an integer with three oscillators and a qubit

Corresponding Author: Dr Libor Caha

Version 0:

Reviewer comments:

Reviewer #1

(Remarks to the Author)

This work proposes a new approach to the construction of quantum algorithms which is more closely connected to the underlying physical hardware and the operations available with this hardware. The authors apply this concept in detail to a variant of a factoring algorithm reminiscent of Shor's algorithm, but with significant modifications based on the above concept. Unlike for the original Shor algorithm, where both the number of input qubits and the number of gate operations grow with the size of the number to be factored, the authors show that a constant number of initial quantum states – three quantum oscillators and one qubit – suffice, and that a polynomial increase of the number of elementary gate operations alone works in principle.

The obvious reason for this is that even a single, infinite-dimensional oscillator can carry and process a lot of information, whereas the encoding of the relevant large numbers in Shor's original scheme requires enough qubits. Less obvious and conceptually even more interesting is how to encode those numbers into the oscillators and how to process them in a suitable way using physically available interactions as provided by common quantum hardware platforms. For this purpose, the authors propose to use Gottesman-Kitaev-Preskill (GKP) oscillator states which are represented by lattices in phase space, and which are most commonly (and originally) associated with GKP qubits, i.e., two-dimensional subspaces of the oscillator space ("encoding a qubit in an oscillator"). GKP qubits exhibit remarkable properties with regards to quantum error correction, fault tolerance, and universality, and these properties motivated their original proposal and much of the subsequent, more recent work related with them. Logical GKP qubits allow to correct small displacement errors into which any physical oscillator error can be decomposed, and they permit efficient deterministic entangling operations which are linear in the mode operators.

Distinct from this, the authors of the present work make use of GKP states to directly encode and process integers through the different peaks in phase space. By physically implementing real arithmetic on GKP states, multiplying and adding numbers by squeezing and displacing the oscillator modes, respectively, the algorithm is directly executed on the physical level. An extra, physical control qubit adds a discrete element (besides the discrete GKP-peaks' integers) and allows to decide in a "hybrid" qubit-oscillator gate whether, for instance, a multiplication should take place or not. The corresponding qubit-oscillator gates are physically available in platforms that include Jaynes-Cummings-type interactions such as solid-state circuit-QED or atomic cavity-QED.

This work is a significant contribution to the field of quantum computing. It presents a new, promising concept for an important, prominent application of quantum computing and explicitly works it out in detail and with mathematical rigor. Unlike some other schemes related with "quantum supremacy", the current scheme is for solving a problem of practical interest. Elements of the proposal have been known and used before, such as hybrid qubit-oscillator gates and GKP states (for both of which universality proofs exist) but combining them in this context and for this type of application is a new idea. The significance of this idea is that it appears to greatly reduce the necessary resources required in a quantum algorithm.

The paper is very well written. The structure is clear and concise, presenting the most important steps and elements in the main text, and the technical details in the appendix. The detailed figures in the main text further help to illustrate the overall concept. The authors explain their scheme in a "modular way", and for some of the modules (such as unitary blocks), they first discuss the ideal circuits with unphysical GKP states and as a second step the real circuits with physical states. The treatment of the circuits with physical states is presented in detail in the appendix. This proposal is still a "proof-of-principle proposal" (Conclusions). To exploit the oscillator Hilbert space, the GKP "squeezing" grows with the number to be factored, corresponding to an increase of the average excitation number of the oscillator states. Such high-quality GKP states are not

easy to build in any platform. In addition, there is no analysis of fault tolerance (which may be naturally embedded via GKP error correction). These subtleties are openly discussed in the Conclusions section. Therefore, based on the discussion above, I recommend this paper for publication in Nature Communications. There are only a few comments and suggestions that may help to improve the manuscript, which are listed below.

-Terminology: the term boson and bosonic code/state is sometimes used in a slightly misleading way, line 37: "...the underlying physical system (3 bosons and 1 qubit) fixed...". It's better to strictly stick to the terminology of the paper title with "3 oscillators and 1 qubit". Each oscillator in a sufficiently squeezed GKP state will carry many bosons. (What also works is "3 bosonic states/modes and 1 qubit".)

-Notation: N is the given integer to be factored, n is the number of bits to represent N ; in line 67, the binary representation of the integer x uses a little m ($0 \dots 2^m - 1$) in the definition of the pseudomodular power function. Is a new parameter m really needed here? Could we just use n again? Moreover, in line 75, it is again referred to an n in " $O(n^2)$ elementary operations".

-Success Probability: in line 57 and in line 92 of Theorem 1, a "constant probability" is mentioned for finding a factor of N . Can we say something about the size of this constant probability (it may be mentioned in the appendix, but it may also be good to say something about this in the main text).

-Caption of Fig.1: "The two-mode bosonic addition gate... can be decomposed into constantly many beam splitters and single-mode squeezing unitaries [3]." It is well-known from the fields of continuous-variable quantum computing and quantum optics that this gate, also referred to as a CV CSUM gate, can be, like any two-mode squeezing gate, decomposed into a beam splitter, two single-mode squeezers, and another beam splitter ("Bloch-Messiah reduction"). This could simply be stated here, possibly including relevant citations (generally, throughout the paper, there is a little bit of a lack of citations of works related with continuous-variable quantum computing and optics; many of these other works originated from the optics/photonics community, but nonetheless, many of these results directly apply here as well; the primary citation used, the more recent Ref.[3], is already adapted to the hybrid qubit-oscillator setting, but the pure oscillator gates have been introduced and used to a great extent already before in the context of CV QC; there are also earlier "hybrid qubit-oscillator papers", but they may be cited in Ref.[3]).

Reviewer #2

(Remarks to the Author)

This manuscript presents a quantum algorithm for factoring, expressed as a quantum circuit involving an elementary set of operations involving 3 CV systems (oscillators) and a qubit. This is related to, but not identical to, Shor's quantum algorithm for factoring that is expressed in terms of an elementary set of operations on n qubits. This new algorithm provides an alternative way of expressing the resources required to factor numbers using a quantum computer. Because of the keen interest in Shor's algorithm and understanding the resources it requires, this result is noteworthy, as alternative perspectives on the required resources may provide insight into these resource requirements and may lead to some innovation in reducing these resources, although the latter is not explored here.

This result should be of interest to a range of quantum computing researchers, including both those focused on developing and understanding quantum algorithms, and also those (including experimentalists) considering resource-efficient ways of running quantum algorithms. The results in this manuscript are framed using CV quantum operations and GKP states, which are very intensely-studied at the moment, including considerable experimental investigation. Certainly, researchers in CV quantum systems using GKP encodings will be interested in re-casting various quantum algorithms into more 'natural' native operations and exploring if this leads to some efficiency in resources, or even just a different approach to viewing an algorithm or protocol. There is considerable existing research in this broad area and the authors have done a good job at citing the relevant literature.

I am quite confident in the validity of the claims here. There is considerable technical detail given in the appendices, and the authors have summarized the key elements in the main part of the manuscript. I do not see any reason to question the technical correctness of these results, and the work should definitively be published in some form.

This will be an interesting new result for a range of researchers. The remaining question is whether the result is of sufficient importance and significance to merit publication in Nature Communications, based on the journal's selection criteria. Here, I must unfortunately question whether this result will have lasting significance or a large impact. As well summarized by the authors in the conclusions, the paper does not attempt to present any analysis of the resources required, or to compare these resources (in any way) to those needed for Shor's algorithm. They do not touch on the issue of fault tolerance. So the reader is really left wondering, 'what does this mean?' Have we understood something new about the tradeoff in moving from the standard qubit-based approach to the CV approach? Have we gained any understanding on how the range of CV/GKP experimental platforms currently being explored might be able to make use of the results here? I feel that the authors have presented a novel theoretical idea, but it does not yet sit clearly in the broader field of research and there is little here to help guide the reader or give future directions to understand where it sits or might sit.

The manuscript is likely to be more suited for a specialist journal, where researchers in quantum algorithms might access this result over time and start to address some of these questions.

Reviewer #3

(Remarks to the Author)

While the standard model of quantum computation is formulated in terms of discrete variables (DV) in finite-dimensional Hilbert spaces, many degrees of freedom, such as bosons, are continuous variables (CV) described by infinite-dimensional Hilbert spaces. Understanding the power of these degrees of freedom is an important research direction. Due to infinite dimensions, one expects more computational capabilities from such devices. At the same time, due to rapid energy accumulations, controlling CV degrees of freedom is a tedious task, particularly when superquadratic Hamiltonians are considered. An intermediate paradigm of computation is to couple CV and DV degrees of freedom. This paper considers this framework. They use qubit-controlled (up to quadratic) CV operations. Such a gate set includes a quantum Fourier transform over the continuous domain (by measuring a state function that is specified in the position basis in the momentum basis). They furthermore use the well-known GKP states used for quantum error correction as a resource. The main result of this paper is factoring using only three CV degrees of freedom (aka oscillators) and one qubit. The algorithm is very similar to Shor's original algorithm, implemented using native CV operations. This is a neat result. It is furthermore well-written. And I recommend acceptance if there is availability.

The main comment I have for the authors is that, even though the model you use has polynomial "time" complexity and a constant number of modes, it still uses exponential energy. I believe you need to emphasize this early on. You do explain that this is not a practical problem, but emphasizing energy as a resource would be nice as well.

One question I have is: What prevents you from achieving universal quantum computation with this model? You may be aware that follow-up work on Chabaud et al. shows that other CV gate sets using exp energy can solve arbitrary NP problems. Also, you do reference the Shamir result, which uses arithmetic operations in a unit step of time. I believe if we think deeply about that result, the main assumption they are using is also about energy consumption in unit time/space. So, in that regard, it is not immediately clear if a bosonic algorithm like the one you used is fundamentally different. For example, here is a question to think about: Can we implement factoring using a classical CV model of computation?

The result is neat. With the time budget I had I could not verify every detail but they do seem reasonable. What I am struggling to understand is why the supplementary material is very long. After going over the warm-up example, I was relatively convinced that the result should hold for more energy-restricted resources. I recommend offering some insight into the main complications in making everything rigorous. As far as I understand, since the gate set you are working on is Gaussian-coupled to DV degrees of freedom, there are not many tedious domain issues.

The statement of Proposition 1 is a bit unclear to me. What do you mean by suitable in item ii?

We would like to thank the referees for the careful examination of our manuscript and the helpful comments and suggestions in particular. We have now revised the manuscript taking into account the referees' feedback. Please find below our detailed point-by-point responses to the referees' comments along with a summary of the changes made.

Responses to the referees' comments

Reviewer #1 (Remarks to the Author):

This work proposes a new approach to the construction of quantum algorithms which is more closely connected to the underlying physical hardware and the operations available with this hardware. The authors apply this concept in detail to a variant of a factoring algorithm reminiscent of Shor's algorithm, but with significant modifications based on the above concept. Unlike for the original Shor algorithm, where both the number of input qubits and the number of gate operations grow with the size of the number to be factored, the authors show that a constant number of initial quantum states – three quantum oscillators and one qubit – suffice, and that a polynomial increase of the number of elementary gate operations alone works in principle.

Response: We believe this accurately summarizes our contribution.

The obvious reason for this is that even a single, infinite-dimensional oscillator can carry and process a lot of information, whereas the encoding of the relevant large numbers in Shor's original scheme requires enough qubits. Less obvious and conceptually even more interesting is how to encode those numbers into the oscillators and how to process them in a suitable way using physically available interactions as provided by common quantum hardware platforms. For this purpose, the authors propose to use Gottesman-Kitaev-Preskill (GKP) oscillator states which are represented by lattices in phase space, and which are most commonly (and originally) associated with GKP qubits, i.e., two-dimensional subspaces of the oscillator space ("encoding a qubit in an oscillator"). GKP qubits exhibit remarkable properties with regards to quantum error correction, fault tolerance, and universality, and these properties motivated their original proposal and much of the subsequent, more recent work related with them. Logical GKP qubits allow to correct small displacement errors into which any physical oscillator error can be decomposed, and they permit efficient deterministic entangling operations which are linear in the mode operators.

Response: We agree with the need to propose concrete schemes for CV systems: Dimensional considerations alone are not sufficient in order to claim that infinite-dimensional systems are useful for quantum information processing.

Distinct from this, the authors of the present work make use of GKP states to directly encode and process integers through the different peaks in phase space. By physically implementing real arithmetic on GKP states, multiplying and adding numbers by squeezing and displacing the oscillator modes, respectively, the algorithm is directly executed on the physical level. An extra, physical control qubit adds a discrete element (besides the discrete GKP-peaks' integers) and allows to decide in a "hybrid" qubit-oscillator gate whether, for instance, a multiplication should take place or not. The corresponding qubit-oscillator gates are physically available in platforms that include Jaynes-Cummings-type interactions such as solid-state circuit-QED or atomic cavity-QED.

Response: Our proposal to use GKP states in an algorithmic context is indeed new to the best of our knowledge. We also think that the physical realizability of the associated operations in widely available systems is a distinctive feature of our model.

This work is a significant contribution to the field of quantum computing. It presents a new, promising concept for an important, prominent application of quantum computing and explicitly works it out in detail and with mathematical rigor. Unlike some other schemes related with “quantum supremacy”, the current scheme is for solving a problem of practical interest. Elements of the proposal have been known and used before, such as hybrid qubit-oscillator gates and GKP states (for both of which universality proofs exist) but combining them in this context and for this type of application is a new idea. The significance of this idea is that it appears to greatly reduce the necessary resources required in a quantum algorithm.

Response: We thank the reviewer for this positive assessment. Of course, whether or not factoring is a problem of practical interest may be questioned, but we agree that finding computational advantages with potential applications is a central goal for our field. We hope our work inspires novel approaches to quantum algorithms design.

The paper is very well written. The structure is clear and concise, presenting the most important steps and elements in the main text, and the technical details in the appendix. The detailed figures in the main text further help to illustrate the overall concept. The authors explain their scheme in a “modular way”, and for some of the modules (such as unitary blocks), they first discuss the ideal circuits with unphysical GKP states and as a second step the real circuits with physical states. The treatment of the circuits with physical states is presented in detail in the appendix. This proposal is still a “proof-of-principle proposal” (Conclusions). To exploit the oscillator Hilbert space, the GKP “squeezing” grows with the number to be factored, corresponding to an increase of the average excitation number of the oscillator states. Such high-quality GKP states are not easy to build in any platform. In addition, there is no analysis of fault tolerance (which may be naturally embedded via GKP error correction). These subtleties are openly discussed in the Conclusions section. Therefore, based on the discussion above, I recommend this paper for publication in Nature Communications. There are only a few comments and suggestions that may help to improve the manuscript, which are listed below.

Response: We thank the referee for their evaluation and these positive comments. We fully agree with the statement that many key problems remain, including, in particular, the question of fault-tolerance and the required amount of squeezing and associated scalability issues. As we emphasize in the manuscript, our proposal is a proof of principle. It would be exciting to see corresponding small-scale proof of principle experiments.

Reviewer’s comment: *-Terminology: the term boson and bosonic code/state is sometimes used in a slightly misleading way, line 37: “. . . the underlying physical system (3 bosons and 1 qubit) fixed. . .”. It’s better to strictly stick to the terminology of the paper title with “3 oscillators and 1 qubit”. Each oscillator in a sufficiently squeezed GKP state will carry many bosons. (What also works is “3 bosonic states/modes and 1 qubit”).*

Response: We thank the referee for this suggestion. As they correctly point out, the term boson(s) is often used to refer to occupation numbers, and this is not the intended meaning here. Correspondingly, we have adopted this change.

Changes: We changed:

“the underlying physical system (3 bosons and 1 qubit) fixed”

to:

“the underlying physical system (3 oscillators and 1 qubit) fixed”

and also:

“By associating a single physical information carrier (a boson) to each individual qubit, this philosophy emphasizes modularity”

to:

“By associating a single physical information carrier (an oscillator) to each individual qubit, this philosophy emphasizes modularity”

Reviewer’s comment: *-Notation: N is the given integer to be factored, n is the number of bits to represent N ; in line 67, the binary representation of the integer x uses a little m ($0, \dots, 2^m - 1$) in the definition of the pseudomodular power function. Is a new parameter m really needed here? Could we just use n again? Moreover, in line 75, it is again referred to an n in “ $O(n^2)$ elementary operations”.*

Response: In our analysis, we set m equal to $m = 17n$ (see the Supplementary Table 1). The fact that this parameter needs to be different from n is similar to the reason the first register in Shor’s algorithm has size at least $2n$ and not n : This is required to obtain a sufficient precision in the representation of (rational) numbers in the continued fraction algorithm. In other words, m determines the precision. In our case, our concrete choice of $m = 17n$ ensures that our realization of pseudomodular exponentiation is sufficiently accurate even in the non-ideal case when using approximate GKP states.

For this reason, we prefer to keep referring to this parameter by separate variable (instead of e.g., replacing m by $17n$ throughout).

In Line 75, the parameter n is the number of bits used to represent the input of algorithm (i.e., number N to be factored). In other words, it represents the length of the input of the computational problem, and is therefore the natural quantity to measure complexity in.

Changes: We included more precise reference for the parameters of the approximate GKP states used in our algorithm.

Specifically, we have additionally included the following explanation around Line 67:

“ (In our algorithm, we set m to be proportional to n , with a constant to be fixed later, see the Supplementary Table 1).”

and we made the following change around line 75:

“We note that for our choice of parameters (see the supplementary information) ...”

was changed to

“We note that for our choice of parameters (see Supplementary Table 1) ...”

Reviewer’s comment: *-Success Probability: in line 57 and in line 92 of Theorem 1, a “constant probability” is mentioned for finding a factor of N . Can we say something about the size of this constant probability (it may be mentioned in the appendix, but it may also be good to say something about this in the main text).*

Response: Both probabilities can be amplified without change in complexity to an arbitrary constant close to 1 by repetition. The ultimate success probability then depends on the number rounds of repetition in Theorem 1 (i).

Changes: We included the following: Line 57, we added:

“(which can be amplified by repetition to an arbitrary constant without change in complexity)”.

Line 92: we added

“(The success probability can be amplified by repetition or other techniques as in [Shor].)”

right after the Theorem 1.

Reviewer’s comment: *-Caption of Fig.1: “The two-mode bosonic addition gate... can be decomposed into constantly many beam splitters and single-mode squeezing unitaries [3].” It is well-known from the fields of continuous-variable quantum computing and quantum optics that this gate, also referred to as a CV CSUM gate, can be, like any two-mode squeezing gate, decomposed into a beam splitter, two single-mode squeezers, and another beam splitter (“Bloch-Messiah reduction”). This could simply be stated here, possibly including relevant citations (generally, throughout the paper, there is a little bit of a lack of citations of works related with continuous-variable quantum computing and optics; many of these other works originated from the optics/photonics community, but nonetheless, many of these results directly apply here as well; the primary citation used, the more recent Ref.[3], is already adapted to the hybrid qubit-oscillator setting, but the pure oscillator gates have been introduced and used to a great extent already before in the context of CV QC; there are also earlier “hybrid qubit-oscillator papers”, but they may be cited in Ref.[3]).*

Response: We agree that including additional references discussing standard quantum linear optics and its relation to quantum computing make sense.

Changes: We have included a reference for the Bloch-Messiah reduction in the caption of Figure 1.

Additionally, we have included the following new references:

1. Chuang and Yamamoto. Simple quantum computer. Phys. Rev. A, 52:3489–3496, Nov 1995.

2. Lloyd and Braunstein. Quantum computation over continuous variables. *Phys. Rev. Lett.*, 82:1784–1787, Feb 1999.
3. Knill, Laflamme, and Milburn. A scheme for efficient quantum computation with linear optics. *Nature*, 409(6816):46–52, Jan 2001.
4. Kok, Munro, Nemoto, Ralph, Dowling, and Milburn. Linear optical quantum computing with photonic qubits. *Rev. Mod. Phys.*, 79:135–174, Jan 2007.
5. Cerf, Adami, and Kwiat. Optical simulation of quantum logic. *Phys. Rev. A*, 57:R1477–R1480, Mar 1998.

If the reviewer has additional relevant suggestions we kindly ask them to let us know.

Reviewer #2 (Remarks to the Author):

This manuscript presents a quantum algorithm for factoring, expressed as a quantum circuit involving an elementary set of operations involving 3 CV systems (oscillators) and a qubit. This is related to, but not identical to, Shor's quantum algorithm for factoring that is expressed in terms of an elementary set of operations on n qubits. This new algorithm provides an alternative way of expressing the resources required to factor numbers using a quantum computer. Because of the keen interest in Shor's algorithm and understanding the resources it requires, this result is noteworthy, as alternative perspectives on the required resources may provide insight into these resource requirements and may lead to some innovation in reducing these resources, although the latter is not explored here.

Response: As the referee correctly emphasizes, our algorithm is indeed different from Shor's algorithm, or a CV embedding thereof. We share the referee's view that exploring such alternative approaches may lead to new and potentially more easily realizable implementations.

This result should be of interest to a range of quantum computing researchers, including both those focused on developing and understanding quantum algorithms, and also those (including experimentalists) considering resource-efficient ways of running quantum algorithms. The results in this manuscript are framed using CV quantum operations and GKP states, which are very intensely-studied at the moment, including considerable experimental investigation. Certainly, researchers in CV quantum systems using GKP encodings will be interested in re-casting various quantum algorithms into more 'natural' native operations and exploring if this leads to some efficiency in resources, or even just a different approach to viewing an algorithm or protocol. There is considerable existing research in this broad area and the authors have done a good job at citing the relevant literature.

Response: We thank the referee for their assessment concerning the timeliness of our contribution. This is a rapidly evolving subfield of quantum information processing, and it would be great to see more algorithmic applications of improved experimental capabilities.

I am quite confident in the validity of the claims here. There is considerable technical detail given in the appendices, and the authors have summarized the key elements in the main part of the manuscript. I do not see any reason to question the technical correctness of these results, and the work should definitively be published in some form.

This will be an interesting new result for a range of researchers. The remaining question is whether the result is of sufficient importance and significance to merit publication in Nature Communications, based on the journal's selection criteria. Here, I must unfortunately question whether this result will have lasting significance or a large impact. As well summarized by the authors in the conclusions, the paper does not attempt to present any analysis of the resources required, or to compare these resources (in any way) to those needed for Shor's algorithm. They do not touch on the issue of fault tolerance. So the reader is really left wondering, 'what does this mean?' Have we understood something new about the tradeoff in moving from the standard qubit-based approach to the CV approach? Have we gained any understanding on how the range of CV/GKP experimental platforms currently being explored might be able to make use of the results here? I feel that the authors have presented a novel theoretical idea, but it does not yet sit clearly in the broader field of research and there is

little here to help guide the reader or give future directions to understand where it sits or might sit.

Response: The referee raises important open questions. In particular, the issue of fault-tolerance is highly relevant and demands further attention. However, it is also highly non-trivial. We hope our concrete proposal can serve as a paradigmatic example of a protocol for which CV/DV-fault-tolerance tools can be developed.

In terms of resources, our estimates give detailed bounds on the circuit complexity, showing that our algorithm runs in polynomial time (and with polynomial circuit size), similar to Shor's algorithm. A more direct comparison is challenging because the underlying physical setup is very different. We note that the connection between qubit-based computation and CV/DV-circuits has been analyzed further with respect to the amount of energy required, see [Brenner, Dias, Koenig. Trading modes against energy. arXiv:2509.18854]

The manuscript is likely to be more suited for a specialist journal, where researchers in quantum algorithms might access this result over time and start to address some of these questions.

Response: We thank the reviewer for his report. We strongly believe that our work will inspire further research in hardware-inspired quantum computing, hybrid qubit-oscillator architectures, and fault-tolerant protocols for their use. We believe the visibility gained from exposure to the broad readership of Nature Communications will accelerate the uptake of the interdisciplinary questions we raise.

Reviewer #3 (Remarks to the Author):

While the standard model of quantum computation is formulated in terms of discrete variables (DV) in finite-dimensional Hilbert spaces, many degrees of freedom, such as bosons, are continuous variables (CV) described by infinite-dimensional Hilbert spaces. Understanding the power of these degrees of freedom is an important research direction. Due to infinite dimensions, one expects more computational capabilities from such devices. At the same time, due to rapid energy accumulations, controlling CV degrees of freedom is a tedious task, particularly when superquadratic Hamiltonians are considered. An intermediate paradigm of computation is to couple CV and DV degrees of freedom. This paper considers this framework. They use qubit-controlled (up to quadratic) CV operations. Such a gate set includes a quantum Fourier transform over the continuous domain (by measuring a state function that is specified in the position basis in the momentum basis). They furthermore use the well-known GKP states used for quantum error correction as a resource. The main result of this paper is factoring using only three CV degrees of freedom (aka oscillators) and one qubit. The algorithm is very similar to Shor's original algorithm, implemented using native CV operations. This is a neat result. It is furthermore well-written. And I recommend acceptance if there is availability.

Response: We thank the referee for their evaluation and these positive comments, as well as their accurate summary.

We believe the CV/DV-model is of independent interest because it models various physical systems of experimental relevance. We do not see our work as an intermediate step in transitioning to the study of superquadratic Hamiltonians, but rather a different computational model. It will be interesting to see connections.

Reviewer's comment: *The main comment I have for the authors is that, even though the model you use has polynomial "time" complexity and a constant number of modes, it still uses exponential energy. I believe you need to emphasize this early on. You do explain that this is not a practical problem, but emphasizing energy as a resource would be nice as well.*

Response: We have covered this aspect in detail in our discussion section: In fact, about half of this section is devoted to the question of energy/squeezing. While we believe this provides an adequate treatment and would prefer not to complicate the presentation of the algorithm and its complexity earlier in manuscript, we are open to concrete suggestions beyond the corresponding changes made, see below.

Changes: We included the following paragraph right after the statement of the main theorem (Theorem 1) and discussion about the differences between our and Shor's algorithm.

"As discussed further in the Discussion section, our algorithm produces states whose energy scales extensively in N (the number being factored). This is a natural consequence of an embedding of an exponentially large Hilbert space into a system of three oscillators and one qubit."

Reviewer’s comment: *One question I have is: What prevents you from achieving universal quantum computation with this model?*

Response: Universal quantum computation can indeed be achieved with this model, see the work [Brenner, Dias, Koenig. Trading modes against energy. arXiv:2509.18854].

Reviewer’s comment: *You may be aware that follow-up work on Chabaud et al. shows that other CV gate sets using exp energy can solve arbitrary NP problems. Also, you do reference the Shamir result, which uses arithmetic operations in a unit step of time. I believe if we think deeply about that result, the main assumption they are using is also about energy consumption in unit time/space. So, in that regard, it is not immediately clear if a bosonic algorithm like the one you used is fundamentally different. For example, here is a question to think about: Can we implement factoring using a classical CV model of computation?*

Response: The referee raises an intriguing question. In our opinion, however, our result is not directly comparable, neither to the model considered by Chabaud et al., nor the result obtained by Shamir, for different reasons.

The hybrid CV-DV model we consider has the following properties:

1. all elementary operations we use are physically motivated and appear in present experimental setups, for more details see the review [arXiv:2407.10381]. In particular linear optics operations and preparation of a qubit in the $|0\rangle$ state and Hadamard gates are standard. Qubit-controlled displacement and phase space rotations were successfully realized, e.g., [Eickbusch et al. Nat. Phys., 18(12):1464–1469 (2022); Campagne-Ibarcq et al. Nature, 584:368–372 (2020); Boissonneault et al. Phys. Rev. A, 79:013819 (2009)],
2. the energy can grow at most exponentially with the number of elementary operations. In particular, unlike for certain non-Gaussian operations, see, e.g., [Chabaud et al. ’25], our circuits cannot produce states with unbounded energy in a constant time.
3. the complexity (number of elementary operations) is closely related to the (effective) size of a binary representation of any number used during the course of computation (e.g. encoded in the position basis).

These properties are not shared with the computational model used in Chabaud et al. ’25.

For the work by Shamir ’79, the differences to our work are equally stark: Shamir works in a computational model which is entirely detached from any physical realization. Correspondingly, notions such as energy are not naturally defined.

These core differences set our model apart from the ones cited by the reviewer. We believe that these profound differences are not fully captured by a discussion of ‘energy consumption in unit time/space’.

Changes: We have added the following sentence to the Discussion section.

In more recent work, Chabaud et al. [Chabaud et al. ’25] analyze the computational power of a CV systems with superquadratic Hamiltonians. We emphasize

that their model significantly differs from ours as individual gates can increase the energy by a superlinear amount. Such gates appear challenging to realize experimentally at present.

Let us also comment on factoring in a classical CV model: Shamir’s result can be cast as a factoring of integers embedded in a classical CV system but without a physically motivated setup. This distinction is crucial. If one disregards the constraints of a physical implementation, it is trivial to postulate abstract classical models that factor any integer in unit time.

Reviewer’s comment: *The result is neat. With the time budget I had I could not verify every detail but they do seem reasonable. What I am struggling to understand is why the supplementary material is very long. After going over the warm-up example, I was relatively convinced that the result should hold for more energy-restricted resources. I recommend offering some insight into the main complications in making everything rigorous. As far as I understand, since the gate set you are working on is Gaussian-coupled to DV degrees of freedom, there are not many tedious domain issues.*

Response: The complications arise from the following:

- (i) the necessity to decompose the idealized (exponentiation) unitary into elementary operations we use. We give a decomposition whose complexity is better than what is done in Shor’s algorithm, i.e., $O(n^2)$ qubit gates vs. $O(n^2 \log n)$ elementary operations.
- (ii) the necessity to use physically meaningful, that is, normalizable states (instead of formal position eigenstates). The main point of our analysis establishes that our algorithm (initially designed with formal position eigenstates in mind) still approximately evaluates the desired function.

Both giving an efficient decomposition and analyzing the effect of finite squeezing are non-trivial tasks.

Changes: We changed the following in “A circuit composed of physical operations” section:

“The use of physically realizable (approximate) GKP states $|\text{GKP}_{\kappa_A, \Delta_A}\rangle$ and $|\text{GKP}_{\kappa_B, \Delta_B}\rangle$ with (finite) squeezing parameters (κ_A, Δ_A) and (κ_B, Δ_B) in the circuit $\mathcal{Q}_{a,N}$ necessitates a detailed analysis of function evaluation by unitaries on $L^2(\mathbb{R})^{\otimes 3} \otimes \mathbb{C}^2$.”

was changed to

“We give a decomposition of the unitary $U_{a,N,m}$ into elementary operations in the Supplementary Note. We also show this decomposition has the claimed complexity.

The use of physically realizable (approximate) GKP states $|\text{GKP}_{\kappa_A, \Delta_A}\rangle$ and $|\text{GKP}_{\kappa_B, \Delta_B}\rangle$ with (finite) squeezing parameters (κ_A, Δ_A) and (κ_B, Δ_B) in the circuit $\mathcal{Q}_{a,N}$ necessitates a detailed analysis of approximate function evaluation by unitaries on $L^2(\mathbb{R})^{\otimes 3} \otimes \mathbb{C}^2$. This analysis, given in the Supplementary Note, provides detailed estimates on the correctness of the algorithm as a function of the squeezing parameters.”

Reviewer's comment: *The statement of Proposition 1 is a bit unclear to me. What do you mean by suitable in item ii?*

Response: The notion of suitability is defined before the Proposition. We have added a comment referring to this.

Changes: We included “(see Definition 2.1)” that defines a “suitable family of probability density functions”.

Once again, we thank all three reviewers for their constructive comments.